# Swin Transformer Improves the IDH Mutation Status Prediction of Gliomas Free of MRI-Based Tumor Segmentation

**DOI:** 10.3390/jcm11154625

**Published:** 2022-08-08

**Authors:** Jiangfen Wu, Qian Xu, Yiqing Shen, Weidao Chen, Kai Xu, Xian-Rong Qi

**Affiliations:** 1Beijing Key Laboratory of Molecular Pharmaceutics and New Drug Delivery System, School of Pharmaceutical Sciences, Peking University, Beijing 100191, China; 2Academy for Advanced Interdisciplinary Studies, Peking University, Beijing 100191, China; 3InferVision Institute of Research, Infervision, Chaoyang District, Beijing 100025, China; 4Department of Radiology, The Affiliated Hospital of Xuzhou Medical University, Xuzhou 221002, China; 5School of Mathematical Sciences, Shanghai Jiao Tong University, Shanghai 200000, China

**Keywords:** IDH mutation status, Swin transformer, ResNet, image inputs, bounding box

## Abstract

**Background:** Deep learning (DL) could predict isocitrate dehydrogenase (IDH) mutation status from MRIs. Yet, previous work focused on CNNs with refined tumor segmentation. To bridge the gap, this study aimed to evaluate the feasibility of developing a Transformer-based network to predict the IDH mutation status free of refined tumor segmentation. **Methods:** A total of 493 glioma patients were recruited from two independent institutions for model development (TCIA; *N* = 259) and external test (AHXZ; *N* = 234). IDH mutation status was predicted directly from T2 images with a Swin Transformer and conventional ResNet. Furthermore, to investigate the necessity of refined tumor segmentation, seven strategies for the model input image were explored: (i) whole tumor slice; (ii–iii) tumor mask and/or not edema; (iv–vii) tumor bounding box of 0.8, 1.0, 1.2, 1.5 times. Performance comparison was made among the networks of different architectures along with different image input strategies, using area under the curve (AUC) and accuracy (ACC). Finally, to further boost the performance, a hybrid model was built by incorporating the images with clinical features. **Results:** With the seven proposed input strategies, seven Swin Transformer models and seven ResNet models were built, respectively. Based on the seven Swin Transformer models, an averaged AUC of 0.965 (internal test) and 0.842 (external test) were achieved, outperforming 0.922 and 0.805 resulting from the seven ResNet models, respectively. When a bounding box of 1.0 times was used, Swin Transformer (AUC = 0.868, ACC = 80.7%), achieved the best results against the one that used tumor segmentation (Tumor + Edema, AUC = 0.862, ACC = 78.5%). The hybrid model that integrated age and location features into images yielded improved performance (AUC = 0.878, Accuracy = 82.0%) over the model that used images only. **Conclusions:** Swin Transformer outperforms the CNN-based ResNet in IDH prediction. Using bounding box input images benefits the DL networks in IDH prediction and makes the IDH prediction free of refined glioma segmentation feasible.

## 1. Introduction

Glioma is one of the most refractory cancers with a wide range of prognosis, showing a median survival of 14 months for glioblastomas (grade IV) [1] and of more than 7 years for lower grade gliomas (grades II and III) [2]. To evaluate the prognosis and guide individualized treatment, genetic mutation, especially the isocitrate dehydrogenase (IDH) mutation status, is recommended to be the most important marker for glioma diagnostic decision [3]. The new 2021 WHO guidelines even recommend that the first diagnostic delineation relies on IDH-mutation [4]. Clinical studies have found that lower grade gliomas with wildtype IDH were similar to glioblastomas in terms of prognosis [5]. At present, IDH mutation status can only be definitively identified using immunohistochemistry (IHC) or gene sequencing on a tissue specimen, acquired through biopsy or surgical resection. However, three problems hinder an extensive and accurate accessibility to the IDH mutation identification, including the inaccessibility for biopsy or resection before the treatment decision, the unavailability of tumor resection, and the sampling bias of the biopsy tissue [6]. Moreover, IDH mutation status is not static during cancer progression and/or therapy stages. In other words, pathological examinations may be outdated over time and dynamic monitoring is in urgent need. Therefore, a highly efficient, noninvasive, and instant approach for preoperative IDH mutation status prediction is in high demand.

Magnetic Resonance Imaging (MRI) plays a leading role in the non-invasive glioma diagnosis and treatment planning. Vast efforts have been devoted to invasively and preoperatively determine the IDH mutation status from MRI radiographic features [7,8,9,10]. Specifically, indistinct margins and T2-FLAIR mismatch have been verified to be useful in the IDH mutant and IDH wild type differentiation [9]. However, these radiographic features rely on subjective visual assessment of MRI images. It is difficult for the radiologist to distinguish glioma genotypes based on these radiographic features in clinical practice. Fortunately, leveraging the recent advances in machine learning approaches, such as deep learning (DL), SVM, decision tree, etc., IDH mutation status prediction from MRI can be operated accurately and objectively [11,12,13,14,15,16,17]. Among them, DL approaches have received the most notable attention for the reason of their outstanding performance in the molecular biomarker prediction from high-dimensional numeric information or image signal intensities [18,19,20]. Besides IDH prediction [11,12,13,14,15,16,17], DL is also widely applied to 1p/19q [21,22], MGMT [23,24] prediction, etc.

Most of the previous DL studies comprise of two stages. Firstly, the glioma region is manually or automatically segmented along the lesion edge. Subsequently, another classifier is trained to discover abstract task-specific features from the lesion region and predict IDH mutation from these features [18]. However, manual segmentation of the glioma is subjective and time consuming. Training an automatic segmentation network of glioma is also based on the extra manual annotation, and the network performance is highly vulnerable to image quality, which restricts an efficient implementation to real oncology workflow. Additionally, it has been shown that peritumor tissue provided helpful information for diagnosis and prognosis prediction [25,26,27,28]. Therefore, this study hypothesizes that segmenting the glioma lesion subtly on the MRI is not compulsory for the IDH prediction using deep learning. Moreover, almost all the DL studies use classical convolutional neural networks (CNN) to predict IDH mutation status, such as ResNet [11,13,15,17], which is the most wildly used CNN network in IDH prediction. However, new DL architectures, such as Transformer, have been seldom introduced to perform the IDH prediction. Transformer, a novel neural architecture whose empirical performance significantly outperforms the conventional CNNs, can effectively capture long-range contextual relations between image pixels and approach to be a state-of-the-art network for medical image representation [29,30,31,32]. Until now, only one study has applied this framework to IDH mutation status prediction using the TCIA dataset [32], and more research needs to be performed to demonstrate its generalization and compassion to CNNs.

Thus, this study endeavors to build a Transformer-based model to predict the IDH mutation status free of refined tumor segmentation. The following experiments are operated: (i) Transformer-based and CNN-based models are established, respectively. (ii) To evaluate the feasibility of IDH mutation status prediction free of refined tumor segmentation, seven different kinds of image inputs are defined in different rectangle sizes with different amounts of peritumor tissues. (iii) Clinical information relevant to the predictions is added to optimize the model performance. Only T2 images are used for model building in this study, as they are acquired routinely and showed best performance in IDH genotyping [12].

## 2. Materials and Methods

This retrospective study received approval from the ethical review board of Affiliated Hospital of Xuzhou Medical University (AHXZ), Xuzhou, China. The data were anonymous and the requirement for informed consent was waived.

### 2.1. Patients

The data curated from The Cancer Imaging Archive (TCIA, https://www.cancerimagingarchive.net/, accessed on 5 March 2021) was used for model development and internal testing. The patients met the following criteria: (i) pathologically confirmed glioma; (ii) known IDH protein expression; (iii) inclusive preoperative T2 MRI images; (iv) age ≥ 18 years. Corresponding molecular genetic information was obtained from The Cancer Genome Atlas (TCGA) and referred to the previous studies [11,12,21,24]. The list of enrolled patients from TCIA is elaborated in Appendix A.

The external test set was curated from AHXZ, a total of 488 patients who were diagnosed as gliomas (grades II–IV) from January 2015 to December 2020 at AHXZ were considered for inclusion, as shown in Figure 1A. The inclusion criteria were in accordance with the TCIA set and the exclusion criteria were as follows: (i) the absence of IDH protein expression (*N* = 152); (ii) missing preoperative axial T2 images (*N* = 72); (iii) history of brain tumor treatment (*N* = 30).

In a nutshell, the dataset (*N* = 493) used for this study included a cohort from TCIA for model development and internal test (*N* = 259) and another cohort from AHXZ for external test (*N* = 234), as shown in Figure 1B. TCIA IDH mutation status was determined by Sanger sequenced DNA methods and exome sequencing of whole-genome amplified DNA. The AHXZ IDH expression was detected by immunohistochemistry. Additional clinical data of gliomas, including gender, age, and grade distributions, were also collected.

### 2.2. Study Design

The overall study design is summarized in Figure 2. Five key steps were described, including tumor delineation, image processing and augmentation, image inputs definition, network development using different network architectures, and hybrid model development. Ultimately, 16 models were considered for comparison: 7 Swin transformer models with different image inputs strategies, 7 ResNet models with different image inputs strategies, and another 2 hybrid models that integrate images with clinical features.

#### 2.2.1. Tumor Delineation

Using InferScholar (an online research platform supported at https://research.infervision.com/, Beijing, China), the tumor was outlined on the T2 weighted images. Two regions were contoured for each patient from T2 weighted images. The tumor region was masked if it contained necrosis, cyst, or hemorrhage, and the edema region that surrounded the tumor region (if present; note that some patients do not have an edema region) was masked separately. Illustrative examples of the annotated image are shown in Figure 3, where tumor region was marked in red and edema is in cyan. Tumor masks for all the subjects were manually drawn by one neuroradiologist and independently validated by another senior neuroradiologists with more than 10 years of experience in neuroradiology.

Meanwhile, the lesion location information, including location features and hemisphere distribution, was recorded and confirmed. The location features were reviewed on the T2 weighted images by the neuroradiologist based on the pre-defined six location options, namely frontal lobe, temporal lobe, occipital lobe, parietal lobe, others (insula, basal ganglia, thalamus, cerebellum, brainstem) and multiple lobes [33,34]. Spurred by the location features, this research defined one more clinical feature, i.e., hemisphere distribution. Hemisphere distribution includes four categories: left side, right side, both sides, and others (cerebellum and brain stem), which targets to probe that whether the hemispherical information of glioma related to its IDH mutation status.

#### 2.2.2. Imaging Preprocessing and Augmentation

The most commonly used T2 image acquisition parameters were summarized in Appendix A, including MagneticFieldStrength (T), SliceThickness (mm), Manufacturer, and PixelSpacing (mm). MagneticFieldStrength: 3T in the TCIA (42.9%) and in the AHXZ (93.2%), 1.5T in the TCIA (46.1%) and in the AHXZ (6.8%). SliceThickness: 5 mm in the TCIA (61.0%) and 6 mm in the AHXZ (91.1%). Manufacturer: GE in the TCIA (53.7%) and in the AHXZ (86.0%), Philips in the TCIA (11.6%) and in the AHXZ (8.1%), SIMENS only in TCIA (20.8%). PixelSpacing: 0.4–0.5 mm in the TCIA (44.0%) and in the AHXZ (89.8%).

All T2 images were preprocessed sequentially: (i) N4BiasCorrection; (ii) Intensity normalization to zero mean and unit variance; (iii) Selecting the slices that involved the tumor region and discarding the first and the last slices of each case to prevent the slices interference (iv) Resampling to sizes of 256 × 256 and expanding to three channels by simply repeating the first channel. We leveraged the following data augmentations, including geometric transformations and intensity transformations, to improve the model generalization ability. Extra hyper parameters involved in preprocessing and augmentation parameters are detailed in the Appendix A.

#### 2.2.3. Image Inputs Definition

To probe the feasibility of IDH mutation status prediction without refined tumor segmentation, seven different input image strategies were proposed, depending on the proportion of used information about tumor regions, as depicted in Table 1 and Figure 3.

#### 2.2.4. Network Development Using Different Network Architectures

To investigate the superiority of Transformer network in IDH genotyping, we developed the IDH status prediction model using a Swin transformer and CNN-based ResNet architectures, respectively.

The Swin Transformer, a hierarchical vision transformer using shifted windows, was the most popular architecture in tackling computer vision tasks [35]. Since the Swin Transformer has never been used in IDH genotyping in any studies, and more generally biomarker predictions from MRIs, this study forced the Swin Transformer to bridge this gap. Since ResNet has been the most widely used network in the previous studies and performed well in the IDH mutation status prediction [11,13,15,17], this study only used ResNet to build the CNN-based model.

(1)The Swin Transformer network development

The entire classification process and Swin Transformer architecture were illustrated in Figure 4. MRI images inputs (matrix: 256 × 256) were subdivided into non-overlapping 4 × 4 patches, which are then converted into sequences by flattening. Then, linear image embedding was conducted in stage 1 to preserve positional information about the images, and their features were extracted with a Swin Transformer block. In stage 2, a down sampling process was performed on the patch merging layer to merge adjacent 2 × 2 patches into one patch. As the network deepens, hierarchical representations, such as CNN, could be extracted by the Swin Transformer block. A total of four stages were used to generate the final representation. A global average pooling layer was applied to the output feature map in the last stage (i.e., the class token) to perform the Classification Head, then a linear classifier output the prediction. The Swin Transformer block was also displayed in Figure 4 and detailed in Appendix A.

(2)The CNN-based ResNet network development

The conventional CNN-based network was derived from the well-known 101-layer ResNet architecture (i.e., ResNet-101) [36] and initialized using the ImageNet pretrained weights. The ResNet block was displayed in Appendix A.

#### 2.2.5. Hybrid Model Development

Additional learnable fully connected layers were respectively added to the top-performing Swin Transformer or ResNet to build the hybrid network, which used the additional numeric inputs along as complement to the image inputs. Only clinical features indicating significant difference between IDH-mutant and IDH-wild were used for hybrid model building.

### 2.3. Network Implementation

The slices per patient were considered as individual samples in model development and testing, which means that each slice had its own diagnostic probability. For per-patient probability, the mean probability of all the slices was considered. Through the slice-level strategy, the prediction at case level can get rid of the interference of abnormal slice samples, achieving better model generalization. Accordingly, we split the TCIA dataset (*N* = 259, IDHm:112, IDHw:147) into the training&validation set (*N* = 207, IDHm:88, IDHw:119) and internal test set (*N* = 52, IDHm:24, IDHw:28) with a ratio of 8:2. After individual slice sample generation, the training&validation set contains 2668 slice samples (IDHm:1131, IDHw:1537) and the internal test set contains 702 slice samples (IDHm:344, IDHw:358), as illustrated in Figure 1B. The whole AHXY dataset (*N* = 234, IDHm:81, IDHw:153) is used for external test and yields 1119 slice samples (IDHm:332, IDHw:787).

All the models, which were trained with T2 images, were implemented with PyTorch on an Ubuntu 16.04 server using four NVIDIA GeForce RTX 3090 GPU devices. For the Swin transformer, the initial learning rate was set to 1 × 10^−5^ with a batch size of 32 and maximal iteration of 300. For ResNet model, the initial learning rate was set to 1 × 10^−4^ with a batch size of 32 and maximal iterations of 300. EarlyStopping strategy was used for speeding up the training stage, i.e., the training stage was stopped when the loss on train set did not decrease in five epochs. We employed StepLR with default parameters a learning rate scheduler. Adam optimizer was used for network optimization with β_1_ = 0.9 and β_2_ = 0.99.

### 2.4. Statistical Analysis

The statistical analysis was performed on SPSS 26.0 with *p* < 0.05 considered significant. Continuous variables were expressed as means with corresponding standard deviation and categorical variables were described as proportions. Continuous variables were compared using the Mann–Whitney U test for non-normally distributed and differences in categorical variables were assessed by the chi-squared test or Fisher’s exact test between the train set and the test set as well as between the IDH-mutant and the IDH-wild groups.

Receiver operating characteristic curve (ROC) analysis was performed to obtain the area under the curve (AUC). The probability threshold for the accuracy (ACC) calculation was set to 0.5, thus a predicted probability of ≥0.5 was classified as an IDH-mutant, and other values were classified as IDH-wild. The diagnostic probability per patient was measured by the mean probability from all involved individual slices.

## 3. Results

### 3.1. Patient Data

As shown in Table 2, in terms of IDH status, gender, age, and WHO grade, the AHXZ set had no difference compared with the TCIA set with *p* = 0.053, *p* = 0.277, *p* = 0.678, *p* = 0.059, respectively. While a significant difference was found in location features (*p* < 0.05) and hemisphere distribution (*p* = 0.01).

According to the statistical results between IDH-mutant and IDH-wild in the two datasets, the age of IDH-wild was significantly higher than that of IDH-mutant in both TCIA set (*p* < 0.05) and AHXZ set (*p* < 0.05). Location features (*p* < 0.05, *p* = 0.002) and WHO grade (*p* < 0.05, *p* < 0.05) also yielded significant difference in TCIA set and AHXZ set. No significant difference was found in gender and hemisphere distribution.

Lastly, only age and location features were reserved for hybrid model development. We did not use WHO grade in the hybrid model building because this data remained unknown prior to the surgery.

### 3.2. Performance of the Models with Different Architectures and Input Image Strategies

The results of the Swin Transformer and ResNet model on both the TCIA internal test set and the AHXZ external test set were summarized in Table 3. Only the patient-level results were displayed in Table 3 and the corresponded slice-level results were supplemented in Table 2.

With the seven proposed image input strategies, seven Swin Transformers and seven ResNet models were built, respectively. The seven Swin Transformer models obtained an average internal test AUC, internal test ACC, external test AUC and external test ACC of 0.965, 92.3%, 0.842, 76.6%, respectively. While these of the ResNet model was 0.922, 89.3%, 0.805, 74.9%, respectively (Table 3). Despite the difference in image inputs, all the transformer models consistently achieved higher AUC than the corresponding ResNets (Figure 5a).

As shown in Figure 5b, the highest AUC (0.984) and ACC (96.2%) for the Swin Transformer in the internal test were obtained using 1.5× Tumor Bbox as inputs, followed by 1.0× Tumor Bbox (AUC = 0.975, ACC = 96.2%), Tumor mask (AUC = 0.975, ACC = 90.4%), 1.2x Tumor Bbox (AUC = 0.965, ACC = 92.3%), Tumor Slice (0.955, 90.4%), 0.8× Tumor Bbox (AUC = 0.953, ACC = 88.5%), Tumor + Edema (AUC = 0.946, ACC = 92.3%). While in the external test set, the Swin Transformer with 1.0× Tumor Bbox yielded the highest AUC = 0.868 and ACC = 80.7%, followed by refined tumor segmentation (Tumor + Edema, AUC = 0.862, ACC = 78.5%). Similar results were found in the ResNet models; the ResNet model obtained the best AUC and ACC using the 1.0× Tumor Bbox in both the internal test set (AUC = 0.938, ACC = 92.3%) and external test set (AUC = 0.831, ACC = 77.3%), followed by tumor mask in the internal test set (AUC = 0.936, ACC = 90.4%) and Tumor + Edema in the external test set (AUC = 0.823, ACC = 76.8%). In general, using 1.0× Tumor Bbox as inputs, both the Swin Transformer and ResNets achieved the best performance on the external test, slightly superior to the models that used refined segmentation of Tumor + Edema.

### 3.3. Performance of the Hybrid Model

According to the above results, we built the hybrid model with the 1.0× Tumor Bbox as image input. Besides the image input, age and location information was also used as input in the hybrid model. Compared to the image-based models, the hybrid model achieved similar results on both the Swin Transformer (AUC = 0.975, ACC = 96.2%) and ResNet network (AUC = 0.960, ACC = 93.2%) in the internal test set. While in the external test set, better results were obtained on both hybrid the Swin Transformer (AUC = 0.878, ACC = 82%) and hybrid ResNet (AUC = 0.833, ACC = 78.1%), as shown in Figure 6.

## 4. Discussions

IDH mutation status has great clinical significance and potentially improves the glioma treatment selection. DL approaches built based on MR images are expected to be an efficient alternative to standard invasive biopsy approaches for the IDH status determination and are robust computer-aided diagnostic tools that can be used to assist radiologists. Thus, this study leverages the Swin Transformer as the backbone to tackle three problems in IDH prediction: (1) IDH mutation status forecasting using Transformer backbones rather than CNN. (2) Free of glioma segmentation and consideration of peritumoral tissue. (3) Important clinical information relevant to IDH mutation predictions. Empirically, the Swin Transformer consistently outperformed conventional ResNet models. When 1.0× Tumor Bbox input was used, the Swin Transformer achieved better performance and generalization, compared to that which used refined tumor segmentation (Tumor + Edema). Similar results were observed with ResNet. Furthermore, the hybrid model that combined images and clinical features (age, location feature) as inputs demonstrated performance improvement in the external dataset. To our knowledge, this is the first study using the Swin Transformer network and tumor bounding box to predict IDH mutation status and testing it in an external dataset.

Compared to the previous studies, our top performing image-based Swin Transformer model achieved a robust result in both an internal test set (AUC = 0.975, ACC = 96.2%) and external test set (AUC = 0.868, ACC = 80.7%) in IDH prediction. Two early reported image-based CNN models obtained a comparable high accuracy of 0.94–0.97 in the internal public dataset [12,14] without performing external testing in a separate dataset. There were also two image-based studies that performed external testing [11,15]. Choi [11] performed IDH mutation prediction of AUC = 0.81 and ACC = 73.5% in the external testing using multimodal images as inputs rather than single T2 images. In Ken’s study [15], T2 image-based external testing achieved AUC = 0.73 and ACC = 67.3%, which was inferior to our results. Besides the CNN-based studies, only one study introduced the transformer to the IDH genotyping [32] without external testing, achieving an internal TCIA test of AUC = 91.04% and ACC = 90%, which was lower than our internal test results. Our Swin Transformer network with bounding box inputs showed great potential in IDH mutation prediction.

The Swin Transformer yielded overall better performance than the ResNet, consistently with the same image inputs strategies. Since IDH expression showed no significant signs on the conventional MR images, it was a great challenge to improve the feature learning efficiency on the DL model. Three structures contributed to the Swin Transformer superiority in feature learning and classification: (i) Multi-head self-attention derived from the good noise suppression ability. Specifically, due to the inherent glioma nature of tumor heterogeneity and lesion boundary diffusion, quite a lot of noise mixed with information related to the IDH genotyping. Compared with CNN, the Transformer network was more prudent to the signal noise [37,38,39]. (ii) Hierarchical architecture spurred by the translation invariance advantage of CNN had the flexibility to model at various scales. Although image inputs had a high diversity in image size, the hierarchical architecture enabled the model to capture distinct phenotypic differences on the regional patches as well as the whole lesion. ResNet is good at deep feature representation, while still having limitations in modeling explicit global contexts due to the intrinsic nature locality of convolutional operations [40]. (iii) Shifted windows ensured the global information interaction. The Swin Transformer could effectively capture long-range contextual relations between image pixels while maintaining the low-level feature extraction [35]. In general, the Swin Transformer has a promising future to conduct accurate and robust performance in imaging molecular prediction [40].

To the best of our knowledge, all the previous studies yielded their results in IDH mutation prediction with tumor segmentation as inputs. Different from previous studies, our model pioneered tumor bounding box as inputs and achieved outstanding performance compared to that using tumor mask or larger boxes. Several merits of bounding box inputs deserved discussion. Firstly, it was a paradox to delineate the infiltrating margins of diffuse glioma in a refined manner, while bounding box had higher fault tolerance and reproducibility [25,26]. Secondly, the rectangular frame not only involved the glioma lesion, but also contained the peritumoral area where the tumor microenvironment might provide more information contributing to diagnosis [28]. Thirdly, bounding box drawing was also friendly to the clinical practice. Compared to the elaborate margin drawing, box drawing only relied on the rough lesion position and largely reduced the labor cost for labeling. Moreover, bounding box of 1.0 times was most approaching the IDH mutation status in our results, and we could deduce that this region resection may have survival benefit to the patient [41].

Pretreatment age and location feature could be easily obtained and had good correlation to the IDH mutation status prediction [11,15,33,34]. Our study demonstrated that the age of IDH-mutant gliomas was significantly younger than that of IDH-wild groups on both the TCIA and AHXZ datasets. The location feature results in this study were also in line with the previous studies in which IDH-mutant gliomas occupied a single frontal lobe more frequently, whereas the IDH-wild gliomas predominantly located in multiple lobes [33]. Importing the age and location feature might be a viable option to improve the model performance. However, compared to the image-based model results, our hybrid model obtained little performance improvement in the internal test and slightly better performance in the external test. Two reasons might attribute to this result. Firstly, the image-based DL model performance on the internal test was good enough and even reached the model ability ceiling. Secondly, the location feature distribution between TCIA and AHXZ behaved with significant differences, which weakened its efficacy on the external testing. More research needs to probe the necessity to import the clinical features to the Transformer-based image model.

## 5. Limitations

Several limitations merit discussion. Firstly, this study only focused on T2 images as model inputs, without considering other MR image modalities such as T1 contrast images and diffusion-weighted images (DWI). Given that our goal was to establish a clinical feasible model with the most widely available T2 images, using multi-modality images might limit the model feasibility. Moreover, previous studies indicated that models constructed for T2 images showed better performance than the multi-modality network [12]. Secondly, as a representative of real-world clinical experience, the TCIA data set, with multiparametric MR images from multiple institutions, was friendly to be used to train a model of good robustness. Although the TCIA data set was applied to train the models in this study, only one external test cohort was used, and model generalization to more external datasets need to be assessed. Thirdly, this is a primary study that used the Swin Transformer for IDH genotyping. Therefore, we are looking forward to investigating its further clinical practice by optimizing its structure to enhance model efficiency. Moreover, compared to the Radiomics [42], the interpretability of the Swin Transformer remains to be a challenge and needs further investigation.

## 6. Conclusions

In this research, we developed a robust IDH mutation status prediction model based on T2 weighted images: (i) Swin Transformer overwhelmed the ResNet in predicting IDH mutation status. (ii) Using bounding box input images benefited the Swin Transformer in IDH prediction and made the IDH prediction free of refined glioma segmentation feasible. The Swin Transformer with bounding box input images might have a promising future in clinical practice, facilitating individualized treatment planning.

## Figures and Tables

**Figure 1 jcm-11-04625-f001:**
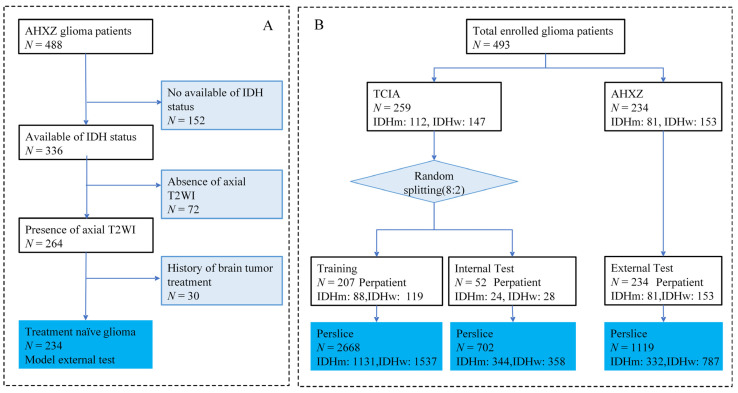
Flowchart of patient enrollment and network implementation. (**A**) Flow diagram of AHXZ dataset enrollment. (**B**) Patient information for model implementation. IDHm = IDH mutant type; IDHw = IDH wild type.

**Figure 2 jcm-11-04625-f002:**
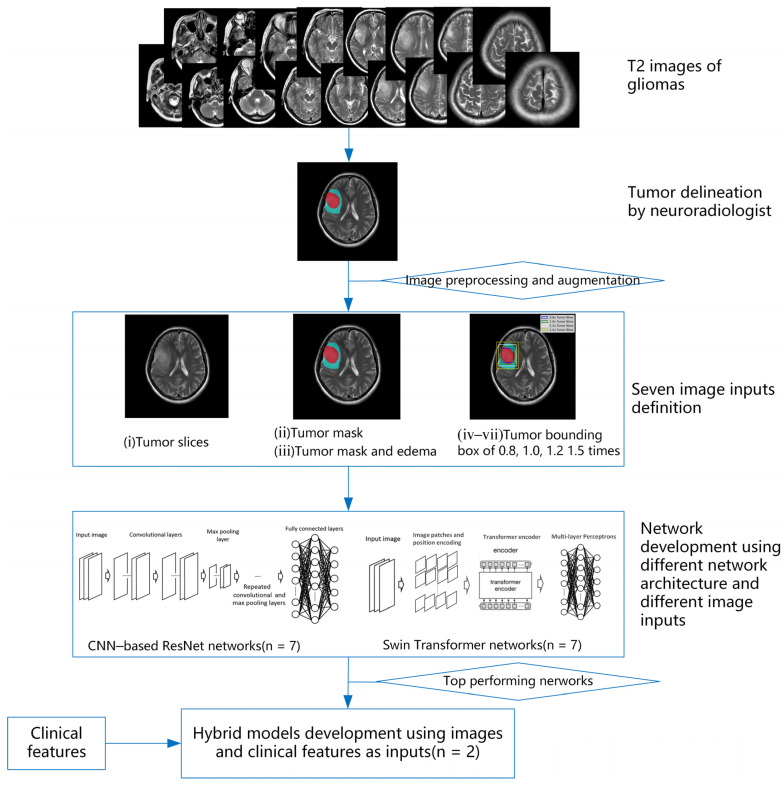
Overview of this study design. This study includes five key steps: tumor delineation, image preprocessing and augmentation, image inputs definition, network development, and hybrid model development.

**Figure 3 jcm-11-04625-f003:**
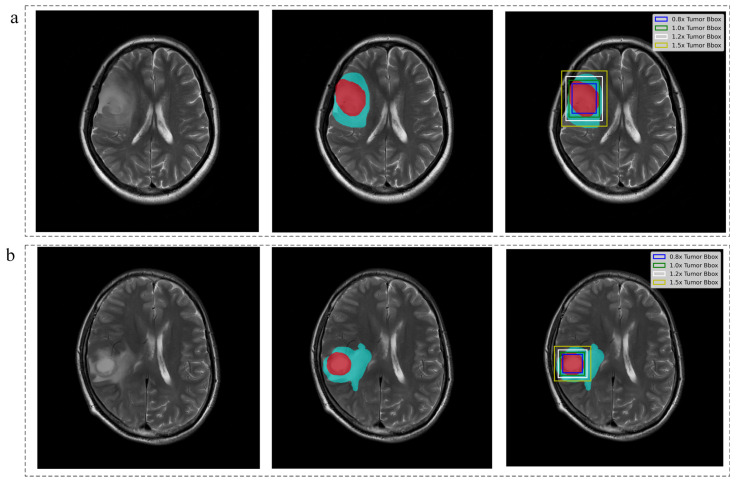
Tumor delineation and image inputs definition. (**a**) IDH-mutant case, female, 59 years old, WHO grade Ⅲ; (**b**) IDH-wild case, male, 53 years old, WHO grade Ⅳ. The two cases enrolled from AHXZ. Red voxels represent tumor and cyan voxels represent edema.

**Figure 4 jcm-11-04625-f004:**
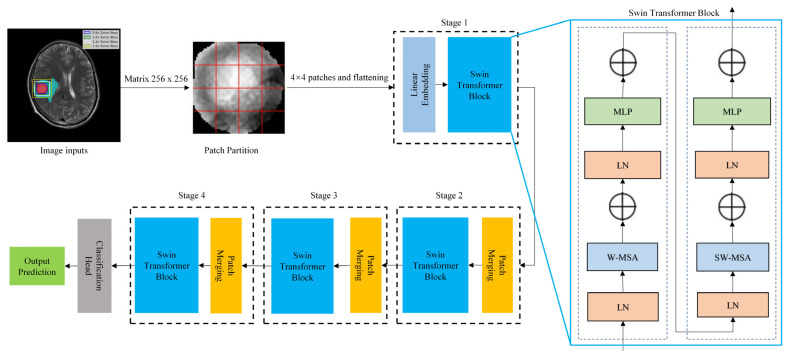
The entire classification process and Swin Transformer architecture. LN: layer normalization; MLP: multilayer perceptron; W-MSA: window multi-head self-attention; SW-MSA: shifted-window multi-head self-attention.

**Figure 5 jcm-11-04625-f005:**
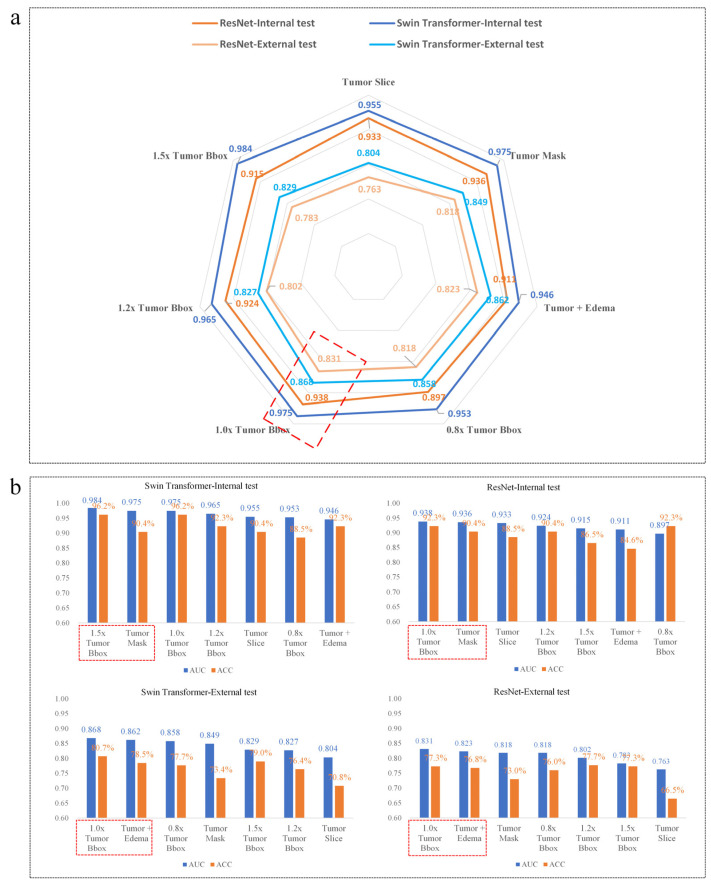
Visualization of model performance. (**a**) Radar map of the model AUC. When the same input images were used, the transformer models obtained higher AUC than the corresponding ResNet models. (**b**) Histogram of the model AUC and accuracy (ACC). All the histogram was presented in the order of model AUC from highest to lowest. Using 1.0× Tumor Bbox images as inputs, both the Swin Transformer and ResNet models achieved the best performance on the external test.

**Figure 6 jcm-11-04625-f006:**
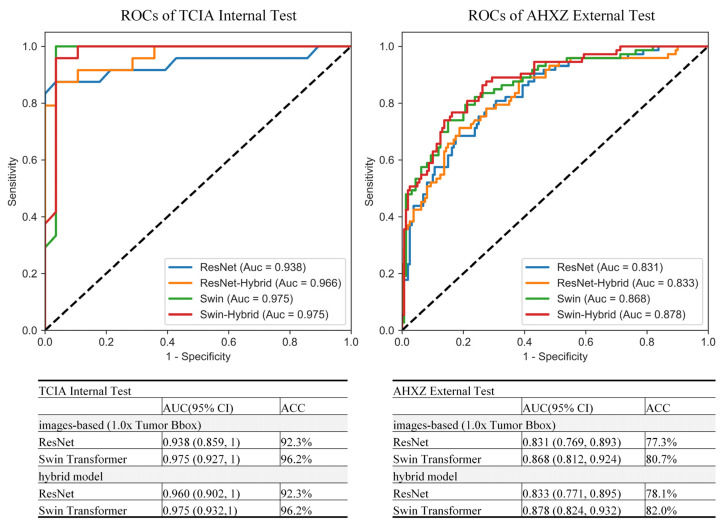
ROCs of the top performing image-based models and their corresponding hybrid models.

**Table 1 jcm-11-04625-t001:** Description of the seven pre-processing strategies for the input images.

	Image Inputs	Description
**i**	Tumor slice	the whole slices contained the tumor mask
**ii**	Tumor mask	the tumor region alone by setting all outside tumor pixels as zero
**iii**	Tumor mask + Edema	The joint region that contained both tumor region and the edema region by setting all outside pixels as zero
**iv**	0.8× Tumor Bbox	downscaled the bounding box of tumor mask by 0.8
**v**	1.0× Tumor Bbox	bounding box of tumor mask
**vi**	1.2× Tumor Bbox	enlarged the bounding box of tumor mask by 1.2 times
**vii**	1.5× Tumor Bbox	enlarged the bounding box of tumor mask by 1.5 times

Note: both Tumor mask and Tumor mask + Edema were defined as refined segmentation input image; The 0.8× Tumor Bbox, 1.0× Tumor Bbox, 1.2× Tumor Bbox and 1.5× Tumor Bbox were defined as free of segmentation input image.

**Table 2 jcm-11-04625-t002:** Patient characteristics.

	TCIA (*N* = 259)		TCIA *p*-Value (IDHm vs. IDHw)	AHXZ (*N* = 234)		AHXZ *p*-Value (IDHm vs. IDHw)	*p*-Value (TCIA vs. AHXZ)
IDHstatus	IDH-mutant	IDH-wild	-	IDH-mutant	IDH-wild	-	0.053
	112(43.2%)	147(56.8%)		81(34.6%)	153(65.4%)		
gender			0.208			>0.999	0.277
female	58	63		34	64		
male	54	83		47	89		
age	51.5 ± 15.7		<0.05	52.2 ± 13.1		<0.05	0.678
	42.4 ± 13.8	58.5 ± 13.3		47.22 ± 11.7	54.79 ± 13.1		
locationfeatures			<0.05			0.002	<0.05
frontal	53	33		36	31		
temporal	18	42		9	16		
occipital	1	4			1		
parietal	13	24		1	10		
Lociothers ^#^	6	3		5	13		
Multiplelobes	21	41		30	82		
hemispheredistribution			0.924			0.097	0.01
left	53	73		44	61		
right	53	67		28	64		
bothsides	6	7		9	23		
hemisphereothers ^##^					5		
WHOgrade			<0.05			<0.05	0.059
2	62	6		40	23		
3	44	23		24	17		
4	6	118		17	113		

Note: IDHm = IDH mutant type; IDHw = IDH wild type. ^#^ Loci Others including Insula, basal ganglia, thalamus, cerebellum, brainstem; ^##^ hemisphere others including cerebellum and brain stem.

**Table 3 jcm-11-04625-t003:** Patient-level diagnostic performance of the models for the IDH mutation status prediction.

	TCIA Internal Test Set		AHXZ External Test Set	
	AUC	ACC	AUC	ACC
ResNet				
Tumor Slice	0.933	88.5%	0.763	66.5%
Tumor Mask	0.936	90.4%	0.818	73.0%
Tumor + Edema	0.911	84.6%	0.823	76.8%
0.8× Tumor Bbox	0.897	92.3%	0.818	76.0%
1.0× Tumor Bbox	0.938	92.3%	0.831	77.3%
1.2× Tumor Bbox	0.924	90.4%	0.802	77.7%
1.5× Tumor Bbox	0.915	86.5%	0.783	77.3%
average	0.922	89.3%	0.805	74.9%
Swin Transformer				
Tumor Slice	0.955	90.4%	0.804	70.8%
Tumor Mask	0.975	90.4%	0.849	73.4%
Tumor + Edema	0.946	92.3%	0.862	78.5%
0.8× Tumor Bbox	0.953	88.5%	0.858	77.7%
1.0× Tumor Bbox	0.975	96.2%	0.868	80.7%
1.2× Tumor Bbox	0.965	92.3%	0.827	76.4%
1.5× Tumor Bbox	0.984	96.2%	0.829	79.0%
average	0.965	92.3%	0.842	76.6%

Note: AUC = area under the ROC curve; ACC = accuracy.

## Data Availability

The TCIA data presented in this study are available in Supplementary Material Appendix A.

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
