# Peer review of "Swin Transformer Improves the IDH Mutation Status Prediction of Gliomas Free of MRI-Based Tumor Segmentation"

_jcm, 2022, doi:10.3390/jcm11154625_

Round 1

Reviewer 1 Report

See my suggestions in the file.

to be the most important marker for glioma diagnostic decision page 3 line 63

Only T2 images are used for model building in this study, as they are acquired routinely and showed best 119 performance in IDH genotyping [12].

the tumor was outlined on the T2 weighted images

reviewed on the T2 weighted images

456 based on T2 weighted images

The protocol was the same for all patients? The field of MRI was the same? 3Tesla?

what about the size of the lesions? The authors should include this information in the table

INCLUDE IN THE DISCUSSION

machine learning approaches are robust computer-aided diagnostic tools that can be used to assist radiologists

Include this reference

Alves AFF, Miranda JRA, Reis F, de Souza SAS, Alves LLR, Feitoza LM, de Castro JTS, de Pina DR. Inflammatory lesions and brain tumors: is it possible to differentiate them based on texture features in magnetic resonance imaging? J Venom Anim Toxins Incl Trop Dis. 2020 Sep 4;26:e20200011.

Reviewer 2 Report

1. In the materials and methods: Please explain the logic behind selecting 7 types of image inputs. Why 0.8, 1, 1.2, 1.5 have been selected?

2. Although it has been mentioned in a figure legend, please mention the fact that the first and the last slices have been deleted in the text as well. I understand that a neuroradiologist has reviewed each image to ensure that it contains both edema and tumor. If so, please mention it in the text.

3. In the discussion, please add the similarity and differences between your " Swin transformer" model and Radiomics. For example, are the extracted features by Swin transformer interpretable or not?
